Survivorship and feeding preferences among size classes of outplanted sea urchins, Tripneustes gratilla, and possible use as biocontrol for invasive alien algae

Westbrook Charley E. 1 charleye@hawaii.edu
Ringang Rory R. 1
Cantero Sean Michael A. 1
HDAR & TNC Urchin Team
Toonen Robert J. 1
1 Hawaiʻi Institute of Marine Biology, University of Hawaiʻi at Mānoa , USA
Esteban María Ángeles
Electronic publication date: 2015 Sep 15
Publication date: 2015
Volume: 3
Electronic Location ID: e1235
Received 2015 May 7; Accepted 2015 Aug 18
Copyright: © 2015 Westbrook et al.
Copyright year: 2015
Copyright holder: Westbrook et al.
License: This is an open access article distributed under the terms of the Creative Commons Attribution License, which permits unrestricted use, distribution, reproduction and adaptation in any medium and for any purpose provided that it is properly attributed. For attribution, the original author(s), title, publication source (PeerJ) and either DOI or URL of the article must be cited.
License URL: https://creativecommons.org/licenses/by/4.0/

Keywords: Hawaiʻi, Kāneʻohe Bay, Acanthophora spicifera, Gracilaria salicornia, Eucheuma denticulatum, Kappaphycus clade B, Super sucker, Kappaphycus striatum, Eucheuma striatum

Funding: National Science Foundation (NSF) #OCE-0623678 NOAA National Marine Sanctuaries Partnership MOA#2005-008/66882 Undergraduate Research Mentoring in Biological Sciences (URM) NSF #0829272 Hawaiʻi Institute of Marine Biology School of Ocean and Earth Science and Technology This project was funded in part by the National Science Foundation (NSF) Grant #OCE-0623678 and NOAA National Marine Sanctuaries Partnership MOA#2005-008/66882 to RJT, RR & SC were supported as research interns under the Undergraduate Research Mentoring in Biological Sciences (URM) Grant (NSF #0829272). This is contribution X from the Hawaiʻi Institute of Marine Biology, Y from the School of Ocean and Earth Science and Technology. The funders had no role in study design, data collection and analysis, decision to publish, or preparation of the manuscript.

==============================
We investigate the survivorship, growth and diet preferences of hatchery-raised juvenile urchins, Tripneustes gratilla, to evaluate the efficacy of their use as biocontrol agents in the efforts to reduce alien invasive algae. In flow-through tanks, we measured urchin growth rates, feeding rates and feeding preferences among diets of the most common invasive algae found in Kāneʻohe Bay, Hawaiʻi: Acanthophora spicifera, Gracilaria salicornia, Eucheuma denticulatum and Kappaphycus clade B. Post-transport survivorship of outplanted urchins was measured in paired open and closed cages in three different reef environments (lagoon, reef flat and reef slope) for a month. Survivorship in closed cages was highest on the reef flat (∼75%), and intermediate in the lagoon and reef slope (∼50%). In contrast, open cages showed similar survivorship on the reef flat and in the lagoon, but only 20% of juvenile urchins survived in open cages placed on the reef slope. Urchins grew significantly faster on diets of G. salicornia (1.58 mm/week ± 0.14 SE) and Kappaphycus clade B (1.69 ± 0.14 mm/wk) than on E. denticulatum (0.97 ± 0.14 mm/wk), with intermediate growth when fed on A. spicifera (1.23 ± 0.11 mm/wk). Interestingly, urchins display size-specific feeding preferences. In non-choice feeding trials, small urchins (17.5–22.5 mm test diameter) consumed G. salicornia fastest (6.08 g/day ± 0.19 SE), with A. spicifera (4.25 ± 0.02 g/day) and Kappaphycus clade B (3.83 ± 0.02 g/day) intermediate, and E. denticulatum was clearly the least consumed (2.32 ± 0.37 g/day). Medium-sized (29.8–43.8 mm) urchins likewise preferentially consumed G. salicornia (12.60 ± 0.08 g/day), with less clear differences among the other species in which E. denticulatum was still consumed least (9.35 ± 0.90 g/day). In contrast, large urchins (45.0–65.0 mm) showed no significant preferences among the different algae species at all (12.43–15.24 g/day). Overall consumption rates in non-choice trials were roughly equal to those in the choice trials, but differences among feeding rates on each species were not predictive of feeding preferences when urchins were presented all four species simultaneously. In the choice feeding trials, both small and medium urchins clearly preferred A. spicifera over all other algae (roughly twice as much consumed as any other species). Again, however, differences were less pronounced among adult urchins, with adults showing a significant preference for A. spicifera and Kappaphycus clade B compared to the other two algal species. These findings indicate that outplanted urchins are surviving on the reef flats and eating a variety of alien invasive algae as intended. Although juvenile urchins show stronger feeding preferences, these animals grow quickly, and adult urchins are more generalist herbivores that consume all four alien invasive algae.

Introduction

Within the last 70 years, Kāneʻohe Bay has become home to many introduced and invasive algal species, whose aggressive growth has smothered corals and overgrown many patch reefs across the bay (Coles, DeFelice & Eldredge, 2002; Conklin & Smith, 2005; Smith et al., 2004; Stimson, Larned & Conklin, 2001; Russell & Balazs, 2009; Bahr, Jokiel & Toonen, 2015). Some of these alien algae species were introduced intentionally, whereas others have unknown origins (are cryptogenic), but appear to have gained a foothold in Kāneʻohe Bay thanks to a combination of reduced grazing intensity and high nutrient influx as a result of sewage discharge into the bay (Stimson, Larned & Conklin, 2001). Among the most obvious and ecologically dominant of these invasive algal species are Kappaphycus clade B, Eucheuma denticulatum, Gracilaria salicornia, and Acanthophora spicifera. The species level taxonomy of Kappaphycus remains a subject of contention (Conklin, Kurihara & Sherwood, 2009; Sherwood et al., 2010). Due to the uncertain nomenclature of Kappaphycus in the literature, it was suggested we use the most contemporary denomination of the alga (despite the prospect of its name changing again, this is the best we could do at the time), henceforth it should and will be referred to as Kappaphycus clade B in this study (K Conklin & A Sherwood, pers. comm., 2015).

Native species of algae which once dominated the Bay (Stimson, Larned & Conklin, 2001; Smith et al., 2004; Conklin & Smith, 2005; Stimson, Cunha & Philippoff, 2007) have become comparatively rare as the rhodophytes K. clade B and G. salicornia both spread rapidly across Kāneʻohe Bay after their introduction, and are now found at high abundance throughout the Bay (Hunter & Evans, 1995; Smith, Hunter & Smith, 2002; Bahr, Jokiel & Toonen, 2015). Originally these alien species were estimated to spread at a minimum rate of 250 m yr−1 (Rodgers & Cox, 1999), although this is now considered to have been a gross underestimate (Coles, DeFelice & Eldredge, 2002; Smith, Hunter & Smith, 2002). Since its introduction and release in 1974, K. clade B has resulted in phase-shifts across the bay by replacing native algae and corals with newly formed monocultures of this alien alga over large areas of reef flat and slope (Coles, DeFelice & Eldredge, 2002; Smith, Hunter & Smith, 2002). Likewise, alien algal overgrowth is smothering live corals on patch reefs, resulting in a loss of biodiversity, changes in community structure of the reef fishes, and erosion of the physical structure of the reef (Smith, Hunter & Smith, 2002). In response to the spread of ecological impacts associated with these alien invasive species, the State of Hawaiʻi Division of Aquatic Resources (DAR) and The Nature Conservancy (TNC) have undertaken manual removal efforts using suction-assisted divers supported by the “supersucker” barges. The “supersucker” teams selectively remove invasive algae species from reefs with sorters on the surface looking for any native species accidentally removed from the reef. Materials that are sucked up through the pumps are sorted on the deck of the barge and any non-alien algae species are returned to the reef immediately. The alien algae is bagged and given to organic farmers who use it as natural fertilizer. However, removal of alien algae by these teams is labor-intensive and only effective if the algae do not regrow, which was happening within roughly a year in the initial supersucker trials (DAR & TNC, pers. comm., 2014). Thus, the long-term solution proposed for areas where invasive algae have begun to overgrow and smother native corals has been to increase the population of native herbivores such as grazing urchins (Conklin & Smith, 2005; Stimson, Cunha & Philippoff, 2007; Weis & Butler, 2009).

Biocontrol agents have been the topic of much debate due to infamous cases during which their introduction lead to their subsequent invasion (Howarth, 1983; Simberloff & Stiling, 1996). Introduced biolcontrol agents, which turned invasive, have wreaked irreversible damage to many host ecosystems (Howarth, 1991). Notable instances of failed biocontrol efforts in Hawaiʻi include the introduction of the Rosy Wolf Snail (Howarth, 1985; Holland, Taylor & Sugiura, 2012), as well as the Indian Mongoose (Simberloff et al., 2000; Godwin, Rodgers & Jokiel, 2006). Indeed, there are precious few examples of successful biocontrol efforts with alien species (Howarth, 1983; Godwin, Rodgers & Jokiel, 2006). Many now argue that if biocontrol agents are to be used at all, they should be native to the ecosystem being targeted (Howarth, 1985). In addition to eliminating the likelihood that alien biocontrol agents become pests in a novel environment, it has been documented that native grazers suppress the establishment of exotic plants better than the introduction of exotic grazers (Parker, Burkepile & Hay, 2006; Kimbro, Cheng & Grosholz, 2013).

Following on that logic, experiments with native sea urchins have demonstrated that T. gratilla have reduced the biomass of the invasive Kappaphycus spp. within enclosure areas on the reef where alien algae were abundant (Conklin & Smith, 2005). Urchins are an important part of the macro-grazing fauna on many tropical reefs, including those in Hawaiʻi (Chiappone et al., 2002; Alves et al., 2003; Mumby et al., 2006; Stimson, Cunha & Philippoff, 2007; Sandin, Walsh & Jackson, 2010; Valentine & Edgar, 2010). Although some urchins show dietary preferences in laboratory studies (e.g., Stimson, Cunha & Philippoff, 2007; Seymour et al., 2013), others appear to be generalist herbivores that will graze on just about any algae or sea grass made available (e.g., Vaïtilington, Rasolofonirina & Jangoux, 2003; Dworjanyn, Pirozzi & Wenshan, 2007). Other potential biocontrol agents, such as fish (acanthurids and scarids) exhibit a relatively low degree of preference for the invasive algae, are far more motile, and are highly prized by local fishermen, making it difficult to rely on herbivorous fishes as a potential mechanism of biocontrol (Conklin & Smith, 2005). Urchins therefore make an obvious choice for a variety of algal biocontrol efforts because of their generalist feeding behavior and limited vagility as adults, coupled with the high dispersal potential and the ubiquity of habitat that has allowed a number of tropical urchins to successfully colonize reefs across the globe (Lessios, Kane & Robertson, 2003; Seymour et al., 2013). Likewise, T. gratilla was historically abundant in Kāneʻohe Bay (Ogden, Ogden & Abbott, 1989; Thomas, 1994), but for unknown reasons has become rare since the 1990s (Stimson, Cunha & Philippoff, 2007; Bahr, Jokiel & Toonen, 2015).

Studies on the ecological impacts of natural outbreaks of T. gratilla corroborated the efficiency with which these urchins can significantly reduce the abundance of algae (Valentine & Edgar, 2010). Due to their limited movement as adults and their voracious appetite for a wide variety of algae and seagrasses, T. gratilla has been recommended as the best species for use as a biological control agent in Kāneʻohe Bay (Conklin & Smith, 2005; Stimson, Cunha & Philippoff, 2007). Since 2010, DAR has been culturing juvenile T. gratilla for outplanting as herbivorous biocontrol agents to prevent regrowth of algal biomass once the alien algae have been manually removed from patch reefs in Kāneʻohe Bay (Gibo, Letsom & Westbrook, 2012). This study set out to investigate if tank bred urchins would eat the targeted alien algae species, and if so, to determine their potential grazing rates. The project also examined if the urchins’ growth could be sustained on diets of non-native algae, and to what extent each alga facilitated growth of T. gratilla. Potential feeding preferences between the four alien algae were also evaluated. Lastly, urchins were caged in various habitats in order to elucidate post-transplant survival of tank bred juvenile urchins in the bay.

Currently, thousands of cultured urchins are outplanted at 20–25 mm test diameter, but comparatively few were observed in subsequent surveys of urchin density on the reef (J Blodgett, DAR, pers. comm., 2014). A major motivation of this research was that it was unknown at that time whether the missing urchins were dying from transplant stress, starvation, being eaten by predators after outplanting, or simply moving into cryptic habitats at small sizes such that they were missing in subsequent surveys. Stimson, Cunha & Philippoff (2007) conducted feeding preference trials with large T. gratilla (8–9 cm) and showed that feeding preferences were generally unchanged after 5 months on monospecific diets, except urchins that were maintained on Padina sanctae-crucis and showed enhanced preference in subsequent choice trials, whereas those maintained on G. salicornia tended to avoid it when offered five species from which to choose at the end of the trial. Further, Stimson, Cunha & Philippoff (2007) found that urchins offered a variety of algal species consume more per day than when limited to a single-species diet. This study expands on the previous work to elucidate patterns of the post-transport juvenile urchin survival, growth and diet preferences of lab cultured T. gratilla being outplanted in Kāneʻohe Bay. Together, these studies will aid both State and conservation group efforts to control alien algal overgrowth of corals on reefs in Kāneʻohe Bay and across Hawaiʻi.

Materials and Methods

Study animal

The short-spined collector urchin Tripneustes gratilla received its common name from the habit of gathering fragments of coral rubble, rocks, or algae from the benthic environment as camouflage while it forages. Its body is predominantly black but often possesses a pentaradial bluish or reddish hue when its tube feet are retracted close to its body. Its spines are typically black, white or cream. This echinoid is relatively common in shallow waters (0–15 m) across the Hawaiian archipelago (Kay, 1994; Hoover, 2002). Natural densities of T. gratilla range from 2.9–4.4 m−2, placing it in the top three most abundant urchins in Hawaiʻi (Ogden, Ogden & Abbott, 1989).

Although T. gratilla have shown significant dietary preferences for Kappaphycus spp. in controlled laboratory studies (Stimson, Cunha & Philippoff, 2007), in the wild these urchins are a generalist herbivore that will graze on virtually any algae or sea grass available (Vaïtilington, Rasolofonirina & Jangoux, 2003; Dworjanyn, Pirozzi & Wenshan, 2007; Stimson, Cunha & Philippoff, 2007). The generalist diet and habitat requirements of T. gratilla coupled with high dispersive potential have resulted in an extremely wide, pantropical distribution (Lessios, Kane & Robertson, 2003). It should be noted that T. gratilla was formerly a native resident of Kāneʻohe Bay. Tripneustes gratilla was once thought to be one of the most abundant urchin species within Kāneʻohe Bay (Edmonson, 1946; Alender, 1964; Banner & Bailey, 1970; Kay, 1994). Conversely, T. gratilla is now relatively rare and does not contribute significantly to herbivory on reefs within the bay (Conklin & Smith, 2005; Stimson & Conklin, 2008). However, historical outbreaks of the native alga Dictyospaeria were thought to be controlled by T. gratilla because growth accumulated mostly in calm waters of the bay where the urchin was rare (Banner & Bailey, 1970). Hence, there is considerable interest from both State and local conservation groups to replenish the natural population of T. gratilla in the bay and enhance natural herbivory to control these invasive alien algal species.

Tripneustes gratilla were provided by the DAR urchin hatchery as juveniles. We arbitrarily placed urchins in three non-overlapping size classes: small (17.5–22.5 mm maximum test diameter), medium (29.8–43.8 mm), and large (45.1–65.1 mm) that were then used for each of the experimental trials outlined below. Prior to the experiments, medium and large urchin size classes were raised on diets of all four alien algae. Small urchins were not fed before trials, but instead were placed directly into experiments within a few days of arriving from the hatchery. Therefore, urchins were starved 3–5 days prior to each experiment.

Algae

We chose the four most common species of alien invasive algae found on the patch reefs of Kāneʻohe Bay: Acanthophora spicifera, Gracilaria salicornia, Eucheuma denticulatum, and Kappaphycus clade B. Kappaphycus clade B (formerly identified as Kappaphycus alvarezii, K. striatum, or Eucheuma striatum) and Eucheuma denticulatum were intentionally introduced from the Philippines to Kāneʻohe Bay September 1974 by researchers from the University of Hawaiʻi for scientific research (Doty, 1971; Doty, 1977); fragments apparently drifted away from test sites on the north reef of Moku O Loʻe (Coconut Island) and were also collected and transplanted around the bay by local residents for personal cultivation (Russell, 1983; Batibasaga, Zertuche-González & de San, 2003; Weis & Butler, 2009; Ask et al., 2003). Despite having observed vegetative propagules being released, researchers reported that such propagules were incapable of dispersing over deep water or finding suitable hollows on which to settle (Doty, 1977). This oversight led to the documented proliferation of new eucheumatoid colonies upon the introduction of the algae to test sites around Coconut Island (Doty, 1977). Likewise, the intentional introduction of Gracilaria salicornia by the same researchers to Kāneʻohe Bay occurred September of 1978, specifically for experimental aquaculture aimed at the development of commercial agar production (reviewed by Rodgers & Cox, 1999; Smith et al., 2004). The idea of a commercial agar industry in Hawaiʻi has long since been abandoned, but the introduced G. salicornia has established and spread along the shores of Waikı¯kı¯ and reefs in Kāneʻohe Bay.

In contrast to these intentional introductions, a fragment of Acanthophora spicifera was first documented in Pearl Harbor in the fall of 1952, and was believed to have been transported on the heavily fouled hull of the barge “Yon 146” which was towed to Oʻahu from Guam in 1950 (Doty, 1961). By February 1956, A. spicifera had been documented in Kāneʻohe Bay, making it the first documented accidental introduction to the Bay (Kohn, 1961; Coles, DeFelice & Eldredge, 2002). These invasive macrophytes became not only some of the most dominant benthic organisms, but they have also resulted in the most detrimental impacts to marine communities in the bay (Coles, DeFelice & Eldredge, 2002).

The four algae species had widespread distributions in the bay and were readily available for collection. Acantohophora spicifera and G. salicornia were easily collected nearly anywhere around Coconut Island and around the southern portion of Kāneʻohe Bay. The eucheumoids were consistently collected from patch reefs in the central portion of the Bay. We did not included native algae in this study because their abundance is so reduced in the Bay (Stimson, Larned & Conklin, 2001; Conklin & Smith, 2005; Stimson, Cunha & Philippoff, 2007) that we could not collect enough for this experiment without impacting the remaining population.

Growth on single-species diet

Growth rates of T. gratilla were measured while on single-species diets of each A. spicifera, G. salicornia, E. denticulatum and K. alverezii. For each of the four algal species, three T. gratilla were housed in each of three 15 L replicate aerated flow-through tanks (∼1–2 L/min). In order to monitor individual growth rates of urchins without marking the animals, each tank held a single urchin of each size class: small, medium, and large. For each treatment, algae were provided ad libitum to reduce any resource competition, and all aquariums were cleaned twice a week during which freshly collected algae were provided to each tank. Urchin test diameter was measured using Vernier calipers (VWR) to the nearest tenth of a millimeter each week for a month. An analysis of covariance (ANCOVA) was employed to analyze growth, with initial urchin test size used as the covariate. Tukey’s HSD post-hoc comparison was then performed to determine significance of pairwise differences in average growth rates of urchins on each algae diet (Fig. 1).

Figure 1 Urchin Growth.

Mean weekly growth rate (mm/week ± SE) of Tripneustes gratilla on non-choice diets of algae (Acanthophora spicifera n = 26, Gracilaria salicornia n = 28, Eucheuma denticulatum n = 27, or Kappaphycus clade B n = 9) reared in aquaria over a 4-week period. Note letters identify significant subsets (p < 0.05, Tukey HSD).

No-choice feeding trials

No-choice feeding trials provided juvenile T. gratilla in each treatment with only a single species of alga and measured differences in mean consumption among the treatments with different algal species. For each algal species, two urchins of the same size class were added to each of six replicate tanks (15 L tanks with 1–2 L/min flow rate, as above). Algae were blotted on paper towels to remove excess water and weighed before being placed in each tank. Urchins were allowed to graze for ∼5 days and the amount of remaining algae was weighed as before to calculate the amount of each species consumed per urchin per day. In a few cases, we stopped the experiment after the 4th day because we did not want the urchins to consume more than half of the algae offered in any trial. Consumption rates (grams of algae per day) were then compared by analysis of variance (ANOVA). Tukey’s HSD post-hoc comparison was used to identify significant differences between each of the algae (Fig. 2). The assumptions of Normality and homogeneity of variance of the data were tested using the Shapiro–Wilks test and the Levene’s test, respectively. The null hypothesis for the Shapiro–Wilks test was that the data were Normally distributed; therefore, p-values less than 0.05 suggested that the data were not Normally distributed. For the small size class in the no-choice feeding trials the data were Normally distributed (Shapiro–Wilk, W = 0.96, p = 0.82). The data from the medium cohort from the no-choice trial was also normally distributed (Shapiro–Wilk, W = 0.90, p = 0.28). However, the data from the large urchins of the no-choice feeding trial were only marginally non-normal (Shapiro–Wilk, W = 0.80, p = 0.042). To test the homogeneity of variance among our feeding trial data, the Levene’s test was employed. The null hypothesis for the Levene’s test was that the variances are homogenous. For the urchins in the no-choice feeding trials, the data passed the homogeneity test (Levene’s test, F(3, 44) = 0.56, p = 0.64).

Figure 2 No-choice feeding trials.

Consumption rates (g/day ± SE) by Tripneustes gratilla during non-choice feeding trials of algae (Acanthophora spicifera, Gracilaria salicornia, Eucheuma denticulatum, or Kappaphycus clade B). (A) Small. (B) Medium. (C) Large. Note letters identify significant subsets (p < 0.05 Tukey HSD post-hoc pairwise comparison). For each diet of each size cohort n = 4.

Choice feeding trials

The choice feeding experiment provided all four species of algae, in equivalent amounts, simultaneously to urchins. As with no-choice experiments above, each algal species was blotted and weighed before being introduced to the experimental tanks. For this assay, larger tanks (80 L, ∼4 L/min flow-through) were used to allow room to separate algae into the four quadrants of the experimental tank. Four urchins were then introduced to the middle of the tank and allowed to graze at will for ∼7 days. Again, if any species of alga became low relative to the others (less than half the initial amount), we stopped the experiment a day early to avoid biasing results. At the end of each experiment, algae were removed, blotted and weighed as previously to calculate the amount of algae consumed per urchin per day for each species.

For each choice and no-choice feeding trials, small urchins were provided with ∼100 g of algae initially, whereas medium and large urchins were offered a starting biomass of ∼150 g of algae. To account for any growth or decline of the algae not attributed to urchin grazing during the experiment, both choice and no-choice experiment tanks had a divider such that one half of the tank housed experimental algae and urchins whereas the other side housed only equivalent amounts of algae to serve as a no urchin control. The consumption rates of algae were then calculated as: Consumption =AiACf/ACi−Af

where Ai and Af were the initial and final blotted masses of algae subject to grazing by urchins; while ACi and ACf were the initial and final masses of the algae in the no-urchin control tanks. This equation was used to account for growth of algae over the course of the experiment, but can also account for any unexpected decline in algal biomass unrelated to the grazing trial (Dworjanyn, Pirozzi & Wenshan, 2007; Seymour et al., 2013). Because all species of algae were provided simultaneously during choice feeding trials, the consumption of one species was affected by the consumption of the others, therefore the assumption of independence required to perform an ANOVA was violated. Consequently, choice feeding preference assays were analyzed using a non-parametric Friedman’s rank test, and both parametric and non-parametric analyses are congruent. Relative consumption rates of each algal species were reported (Fig. 3) and ranked. Nevertheless, due to the lack of post-hoc pairwise comparison for Friedman’s rank test, a Tukey’s HSD post-hoc comparison was used to identify significant differences between each of the algae. Again, to test the data’s distribution for Normality and homogeneity of variance the Shapiro–Wilk and Levene’s test were used. The data from the small urchins of the choice feeding trials failed the normality test (Shapiro–Wilk, W = 0.86, p = 0.0036). However, the data for the medium and large sized urchins of the choice feeding trials both passed the normality test (Shapiro–Wilk, W = 0.97, p = 0.27, and W = 0.96, p = 0.17, respectively). For the small urchins of the choice feeding trial, the data passed the homogeneity test (Levene’s test, F(3, 20) = 2.13, p = 0.13). The data from the medium urchins in the choice feeding trial fail to reject the null hypothesis (Levene’s test, F(3, 36) = 0.32, p = 0.81). The data from the cohort of large urchins in the choice feeding trials also passed the test for equality of variances (Levene’s test, F(3, 40) = 1.98, p = 0.13).

Figure 3 Choice feeding trials.

Consumption rates (g/day ± SE) of Tripneustes gratilla during three choice feeding trials of algae (Acanthophora spicifera, Gracilaria Salicornia, Eucheuma denticulatum, and Kappaphycus clade B). (A) Small, n = 6. (B) Medium, n = 10. (C) Large, n = 11.

Field caging experiment

Cages measuring roughly 50 × 50× 75 cm were constructed from 1 cm2 galvanized chicken wire mesh. We constructed both open and closed cages. For closed cages, the mesh extended across all sides including the tops to prevent urchins from being able to crawl out and prevent access by fishes on the reef. In contrast, the sides of the open cages end with back-folded edges and no top to minimize escape of the juvenile urchins from the cage, but still allow open access of predatory fishes. Initial trials with urchins caged in seawater tables indicated that this back-folded edge design (approximating an upside down U) was the most effective for open cages, but urchins still escaped the cages at the rate of 1–2 animals per week. Urchins ranged from 18–22 mm at the start of the experiment. Cages were filled with G. salicornia to provide the juvenile urchins with food and a place to hide, because our initial aquarium trials revealed that urchins were far more likely to escape the open cages in the absence of hiding spots and food in the cage. In the absence of any cover or food, open cages were frequently empty within 24 h in our water table trials (data not shown).

Cages were placed at 6 sites across three habitats, with four cages, three open and one closed control, at each site. 251 urchins were used during these caging experiments. Three habitats surrounding Coconut Island (map in Supplemental Information) were selected to mimic the conditions on the reef to which urchins are being currently outplanted: a protected lagoon, a shallow back-reef and a fore-reef slope each at 1–3 m depth. The protected lagoon had low coral cover, high alien algal cover and minimal water flow, whereas both the back-reef and fore-reef sites had high coral cover, relatively low alien algal cover and relatively high water flow. Each cage was checked three times a week for 30 days to count surviving urchins as well as replenish consumed algae. All studies reported here were conducted under the State of Hawaiʻi, Department of Land and Natural Resources, Division of Aquatic Resources Special Activity Permits sap#2012–63 and SAP#2013–47. Survivorship between treatments was compared using the Kaplan–Meier product-limit method for fitting survivorship curves and comparison by Log-rank (Forsman, Rinkevich & Hunter, 2006), and Wilcoxon non-parametric tests. These statistical tests were done using JMP Pro 11.

Results

Growth on single-species diet

Growth rates of T. gratilla, measured as maximum test diameter (mm), were significantly affected by fixed algal diets (ANCOVA, initial urchin size as covariate, F(5,84) = 10.80, p < 0.001, Fig. 1). Urchins that fed exclusively on diets of either G. salicornia or K. clade B grew at significantly (p = 0.001 and p = 0.009, respectively, Tukey HSD) higher rates (1.58 ± 0.14 and 1.69 ± 0.14 mm/week TD (Test Diameter), respectively) than those urchins that fed on a diet of E. denticulatum. Urchins that fed on a diet of E. denticulatum had the lowest growth rates (0.97 ± 0.14 mm/week TD) out of the four assays, though not significantly lower to urchins on a diet of A. spicifera (1.23 ± 0.11 mm/week TD).

No-choice feeding trials

When presented with no choice, urchins consumed different species of algae at different rates, but the effect varied by urchin size (Fig. 2). On average, large urchins ate G. salicornia at the highest rate of all algal species offered (15.24 ± 0.001 g day−1), and K. clade B at the lowest rate (12.43 ± 1.51 g day−1) although these trends were not significant (ANOVA, F(3,12) = 1.94, p = 1.78, Fig. 2C). Medium sized urchins showed similar trends, but with significant differences in the amounts of algae they consumed on a daily basis (ANOVA, F(3,12) = 8.49, p < 0.05, Fig. 2B). For the medium size class, Gracilaria salicornia was eaten at a significantly higher rate (12.60 ± 0.08 g day−1) than either A. spicifera (10.33 ± 0.36 g day−1) or E. denticulatum (9.35 ± 0.90 g day−1) (p = 0.034 and p = 0.003, respectively, Tukey HSD); K. clade B was also eaten at a higher rate (11.87 ± 0.27 g day−1) than E. denticulatum (p = 0.018, Tukey HSD), but not A. spicifera (p > 0.05, Tukey HSD). Among the small collector urchins, feeding rate patterns were comperable to those of the medium urchins, but with more significant disparities among algal species, (ANOVA, F(3,12) = 51.30, p < 0.001, Fig. 2A). Small urchins offered only G. salicornia had a significantly higher mean consumption rate (6.08 ± 0.19 g day−1) than any other algal assay in the non-choice feeding trial (p < 0.05 for each pairwise comparison, Tukey HSD). Small urchins likewise consumed Eucheuma denticulatum at the lowest rate (2.32 ± 0.39 g day−1), which was significantly lower than both A. spicifera and K. clade B (p < 0.05, Tukey HSD).

Choice feeding trials

Feeding trials in which urchins were offered multiple species of algae simultaneously revealed different patterns than those observed in the non-choice feeding assays. Small urchins significantly preferred to feed on A. spicifera than any of the other three available algae species (Friedman’s rank test, p < 0.05; p ≤ 0.01, Tukey HSD, Fig. 3A). Small urchins did not display any patterns of preference among G. salicornia, E. denticulatum or K. clade B (p > 0.05, Tukey HSD). Likewise medium urchins showed a significant preference (Friedman’s rank tests, p < 0.001, Fig. 3B) for A. spicifera over G. salicornia and K. clade B (p < 0.05 and p < 0.001, respectively, Tukey HSD), as it was consumed at the highest rate (3.69 ± 0.21 g day−1), whereas G. salicornia and E. denticulatum were consumed at intermediate rates (0.95 ± 0.1 and 1.19 ± 0.13 g day−1, respectively). In contrast, medium urchins consumed K. clade B at the lowest rate (0.51 ± 0.10 g day−1). Unlike the small and medium urchins, which avoided K. clade B in the choice trials, large urchins exhibited significant preferences for both A. spicifera and K. clade B (4.30 ± 0.09 g day−1 and 4.31 ± 0.14 g day−1, respectively) in the choice feeding trials (Freidman’s rank test, P < 0.001, Fig. 3C). Large urchins significantly preferred A. spicifera to both G. salicornia and E. denticulatum (p < 0.0001 and p < 0.001, respectively, Tukey HSD). Kappaphycus clade B was also significantly preferred to G. salicornia and E. denticulatum (p < 0.01 and p < 0.05, respectively, Tukey HSD). Whereas G. salicornia and E. denticulatum were consumed at intermediate rates by small and medium urchins, these species tend to be avoided by the large urchins (3.11 ± 0.20 g day−1 and 3.20 ± 0.20 g day−1, respectively) when given a choice of algae on which to feed (Fig. 3).

Caging experiment

Considerable differences in urchin survivorship among the three habitats that they were caged in were found to be significant (Table 1). On the reef flat, 84.4% of urchins deployed in closed cages remained and 75% of urchins remained in the open cages after 29 days (Fig. 4A). Urchins caged in the lagoon had a substantially higher rate of loss than those on the reef flat, with only 56.3% of urchins remaining after 29 days, regardless of cage type (Fig. 4B). After 29 days, 55.2% of urchins remained in closed cages, but only 20% survived in the open cages, giving urchins deployed on the reef slope the highest rate of loss (Fig. 4C).

Figure 4 Caged urchin survivorship.

Survivorship (%) curves reported for urchins deployed in open and closed cages in various underwater habitats for a month. (A) Reef. (B) Lagoon. (C) Reef Slope. Two hundred and fifty-one urchins were used across 6 sites.

Table 1 Analysis of the Kaplan-Meier Survivorship curves for the Urchin Caging Experiment.

A comparison of survivorship distributions between open and closed cages placed in the lagoon, the reef flat and the reef slope.

		df	Log rank	Wilcoxin	Verdict	
			Statistic	P	Statistic	P	
I. Between all treatments		5	47.15	<0.0001	41.36	<0.0001		
II. Between open cages		2	38.62	<0.0001	33.59	<0.0001		
	R vs L	1	4.43	0.0352	3.17	0.0750	R ≥ L	
	R vs S	1	37.79	<0.0001	32.66	<0.0001	R > S	
	S vs L	1	13.01	0.0003	12.06	0.0005	L > S	
III. Between closed cages		2	6.73	0.0346	6.16	0.0460		
	R vs L	1	4.08	0.0434	3.62	0.0571	R ≥ L	
	R vs S	1	6.13	0.0133	5.73	0.0166	R > S	
	S vs L	1	0.05	0.8315	0.15	0.6959	ns	
Notes.

R reef flat

S reef slope

L lagoon

ns not significant

Discussion

Increasing the abundance of native grazers would not only control and remove current alien invasive algae species, but it would also serve to increase the degree of biotic resistance to novel invasive species (Kimbro, Cheng & Grosholz, 2013). Consistent with previous studies (e.g., Lawrence & Agatsuma, 2001; Seymour et al., 2013), Tripneustes gratilla is a generalist herbivore that managed to grow on every species of algae tested, and fed on all algal species offered to them without exception. However, T. gratilla did not interact with every species of algae indiscriminately. The urchins experienced variable growth depending on the diet they fed on. They did not feed on all rhodophytes at the same rate, and even exhibited preferences between the available species when presented with a choice of all four alien algae species. All of the algal diets supported growth, but the algae species that supported the highest growth rates were G. salicornia and K. clade B, whilst urchins that fed solely on E. denticulatum had the lowest growth rates, and A. spicifera sustained intermediate growth (Fig. 1). The reason for differential growth was not investigated, but could possibly be attributed to variation in nutritional content, consumption rate, digestibility or assimilation efficiency (Sterner & Hessen, 1994). For example, growth rates may be higher on G. salicornia than A. spicifera because Gracilaria contains more protein than does Acanthophora (McDermid, Stuercke & Balazs, 2007). Likewise, the low growth rate of urchins on a diet of E. denticulatum could result from the comparatively high dietary fiber of this alga, a compound that resists digestion and lowers assimilation efficiency (McDermid, Stuercke & Haleakala, 2005).

As with consumption rates, feeding preferences were also observed to vary among size classes (Fig. 3). The differences in feeding preference and consumption rates could translate into urchins of different size classes having differential impacts on alien algae, and argue that biocontrol efficiency could be increased by outplanting urchins of the correct size class for the dominant algal species to be controlled. For example, urchins of all size classes consumed A. spicifera preferentially in choice experiments (Fig. 3). Acanthophora spicifera is known to be consumed by other native grazers (Wylie & Paul, 1988; Russell & Balazs, 1994), but the effects of alien algal ingestion on the diets of native herbivores remains unknown (Smith et al., 2004). Our results show a greater disparity of preference among smaller than larger urchins (Fig. 3), suggesting that the role of alien algae in the diets of native grazers may vary ontogenetically. Nevertheless, small urchins showed a significant preference for A. spicifera relative to all other species, whereas larger urchins showed a higher affinity for both A. spicifera and K. clade B. Despite being a preferred species in all choice feeding trials, urchins did not seem to grow at a significantly different rate (1.23 ± 0.11 mm/week test diameter) when fed only A. spicifera relative to any other algal diet (Fig. 1). In fact, growth rates appeared maximal on a diet of G. salicornia which was less preferred in all choice feeding trials (Fig. 3).

Our results contrast those presented by Stimson, Cunha & Philippoff (2007), which focused solely on large adult urchins (7–8 cm test diameter), and showed a significant preference of urchins for Kappaphycus spp. but without including Eucheuma denticulatum in the feeding trials. Here we find that consumption rates of each algal species vary by size class, and we see no significant differences among consumption rates of algal species for urchins in our largest size class (Fig. 2). Given the strong differences seen in diet preference among urchins of different size classes, the discrepancy between our results and those of Stimson, Cunha & Philippoff (2007), could result from a continued change in diet preference and their use of significantly larger urchins than used in our feeding experiments. Among medium urchins, however, significantly more G. salicornia and K. clade B were consumed, and the smallest urchins consumed significantly more G. salicornia than the other three species (Fig. 2). Although the general trend is similar among urchin size classes, the biggest differences among the consumption rates were seen in the smallest urchins, which consume slightly more than twice as much G. salicornia as E. denticulatum (Fig. 2A). However, as urchin test diameter increased, there was a reduction in the difference between feeding assays, until all four species were consumed at statistically indistinguishable rates among adult urchins in the no-choice trials; larger urchins eat more algae, and grazing rates become more homogeneous among the four species of red algae (Fig. 2). Algal palatability can be reduced by increased algal toughness in a range of herbivorous species (e.g., Peters et al., 2002). For some species of echinoderms, it has been documented that larger sizes (body diameter) are associated with greater jaw strength (Ellers & Telford, 1991). Thus, it may be that larger urchins are better able to masticate a wider variety of algae, including species with larger and tougher thalli, such as E. denticulatum and K. clade B.

It is noteworthy that despite the widespread distribution of T. gratilla, caged juvenile collector urchins did not fare equally well in every environment. The caging experiment highlighted significant differences in survivorship among potential outplanting locations (Table 1). Urchins placed in high algal cover, comparatively low-flow lagoonal habitats fared poorly, with only ∼50% survivorship (Fig. 4B). Survivorship was equal for animals in both open and closed cages in the lagoon, and the removal of empty tests (clear evidence of mortality) accounted for all but a couple animals by the end of the month deployment. Thus, we are confident that the decline in urchin number in the lagoon was a result of low survival as opposed to predation or escape. Urchins caged on the reef flat consistently had the highest overall survivorship rates, with nearly 80% of urchins remaining at the end of the experiment. In these sites, we found only a single urchin test in any cage, and we had a few urchins somehow escape the closed cages in our initial water table trials, so we cannot be sure of whether the missing urchins died or escaped, but the high overall survival in both open and closed cages provides evidence that placement of urchins in these habitats is likely to increase the population of herbivorous urchins over time. In contrast, juvenile urchins placed on the reef slope suffered mortality rates as high as 80% in open cages (Fig. 4C). Even in closed cages, survival over the course of the experiment was only slightly above 50%, and again we recovered only 25 total tests in the cages over the course of the experiment. This seems likely to be a result of predation by reef fishes because we see a much more dramatic decline on the deeper water reef slope than in the shallows of the reef flats (Fig. 4). The decline of urchins in the closed cages on the reef slope below what is seen in open cages on the reef flat is somewhat puzzling. However, we noted during the experiment that the saddleback wrasse, Thalassoma duperrey, was particularly abundant around the urchin cages along the reef slope. Wrasses have been documented harassing juvenile T. gratilla urchins (Dafni & Tobo, 1987), and in a couple of cases, fish were even found to have somehow squeezed themselves inside one of the closed cages along the reef slope and were subsequently trapped in the cage on the next day. Although the mesh size was intended to exclude predatory fish, it is possible that wrasses small enough to fit through 1 cm2 mesh may have been responsible for the decline of urchins in closed cages along the reef slope, and the loss of 80% of animals from open cages within a month suggests that outplanting urchins into deeper waters of the bay is simply generating a feeding station.

Conclusion

The estimated overall productivity of these alien invasive eucheumatoids in Kāneʻohe Bay is 20.8 tonnes dry wt/ha/yr, which translates to 5.7 g dry wt/m2/day (Glenn & Doty, 1990). Given this approximation of productivity and the data compiled on consumption rates, a rough estimate of the ideal urchin density can be derived. Here we find that a large urchin (45–65 mm TD) could graze ∼7.5 g of alien algae per day when presented with a mixed diet. This back-of-the-envelope calculation suggests that the grazing rate of one adult urchin/m2 may be just about equal to the predicted growth rates of these algae. Additionally, our caging experiments indicate that mortality rates for juvenile urchins in open cages on the reef flat are on the order of 25%. Thus, a target density of two urchins/m2 is recommended to overcome growth and reduce the biomass of alien algae if urchin grazing is to be effective for biocontrol. Given the high mortality rates for juvenile urchins on the deeper reef slope and the protected lagoon habitats, it is not advisable to invest the effort to culture and outplant juvenile urchins in either environment. Although there is variability in growth rates, juvenile urchins tend to grow quickly and the largest size class of urchins we tested showed no significant preferences among any of the target alien invasive algal species. Given that current efforts by conservation groups aim to manually remove alien algae and then outplant native urchins to the reef flats, where survivorship was high, this study increases confidence that intentional outplanting of juvenile urchins is likely to be an effective means of biocontrol for these invasive alien algae.

Supplemental Information

Supplemental Information 1 Map of Research Sites

Location of caging experiment around Coconut Island in Kāneʻohe Bay. The red star corresponds to our lagoonal site (low water motion). The orange stars represent are caging sites along the reef flat (high water mixing, low predation). And the yellow stars mark caging areas along the reef slope (mixing, higher predation). These three sites were used to assess post transport survival of juvenile urchins which are deployed in the Bay.

Click here for additional data file.

Supplemental Information 2 Growth rates of urchins size cohorts during on no-choice feeding trials

Average growth rates of small, medium and large urchins while on a no-choice diet of either A. spicifera, G. salicornia, E. denticulatum or K. clade B.

Click here for additional data file.

Supplemental Information 3 Urchin growth/feeding/survivorship data

tab 1: Urchin growth data; tab 2: No choice feeding data on 4 different algal diets; tab 3: Choice feeding data with all 4 algal species; tab 4: Urchin caging survivorship data.

Click here for additional data file.

Thank you to Brian Bowen, Mary Hagedorn, Ginnie Carter and Richard Coleman. The members of the HDAR & TNC Urchin Team are David Cohen, Jono Blodgett, Brian Neilson, Andrew Purvus, Frank Mancini, Tristan Walker, Brad Stubbs, Derek LeVault, Cathy Gewecke, Vince Calabrese, Karen Brittain, Neil Rodriguez, Shealin Johnson, Ryan Carr, Hank Lynch, Justin Dennis, and Kirsten Fujitani.

Additional Information and Declarations

Competing Interests

Author Contributions

Field Study Permissions

Robert J. Toonen is an Academic Editor for PeerJ.

Charley E. Westbrook and Robert J. Toonen conceived and designed the experiments, performed the experiments, analyzed the data, contributed reagents/materials/analysis tools, wrote the paper, prepared figures and/or tables, reviewed drafts of the paper.

Rory R. Ringang and Sean Michael A. Cantero conceived and designed the experiments, performed the experiments, contributed reagents/materials/analysis tools, reviewed drafts of the paper.

HDAR & TNC Urchin Team conceived and designed the experiments, contributed reagents/materials/analysis tools, reviewed drafts of the paper.

The following information was supplied relating to field study approvals (i.e., approving body and any reference numbers):

Division of Aquatic Resources Special Activity Permits

SAP#2012-63 and SAP#2013-47.

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
