# Peer review of "Survivorship and feeding preferences among size classes of outplanted sea urchins, Tripneustes gratilla, and possible use as biocontrol for invasive alien algae"

_PeerJ, doi:10.7717/peerj.1235_

## Round 0.1 · original submission · Major Revisions

Please see the attached reviewer comments. We will reconsider your manuscript after a careful revision.

Reviewer 1 ·

Basic reporting

The main question addressed by this study seems to be whether the sea urchin Tripneustes gratilla may be a suitable candidate for biocontrol of four non-native seaweeds in Hawaii. The authors examine this by testing whether the four seaweeds are eaten by the sea urchins and whether the sea urchins survive out-planting. This study is timely and useful as it provides data that is currently lacking in the literature and required to assess the potential effectiveness of the urchin as a biological control.

The introduction does not place the study within a broader context – biocontrol of introduced pest species. Indeed the authors seem to have missed the opportunity to highlight the importance of their use of a native predator to control a non-native pest, particularly given the majority of biocontrol trials aim to introduce non-native predators or parasites to control non-native pest organisms.

The research question is lost in the overly long introduction. For example, there are several paragraphs which outline how the non-native algae came to be established in Hawaii. While interesting, this information is of little relevance. Instead it is most relevant for information to be briefly presented on how widespread the seaweeds are, the ecological or other impacts of the seaweeds and, especially in the context of this study, the occurrence of native seaweeds alongside the non-natives. More information on “supersucker” barges could be provided to enable the reader to understand the context within which the experiments were conducted, especially given the relevant references cited by the authors appear to be predominately non-peer-reviewed ‘grey literature’. Most notably, do the “supersucker” barges remove all seaweeds (native and non-native if present)? If native seaweeds are present and not removed by the “supersucker” barges, why weren’t native seaweeds also tested alongside the invasive seaweeds in the feeding preference experiments?

The data figures are crude and difficult to read. Simplify. Use both vertical and horizontal axes and remove the horizontal lines behind the bars. The bars should be labelled underneath instead of using different colours/patterns and a legend.

Figure 3 - The results of the statistical analyses for the choice feeding trials should be presented on the figure.

Move Figure 4 to Supplementary Material.

Figure 5 – Error bars should be added.

Experimental design

Information should be given in the methods about how the sea urchins were cultured. This information is important in understanding and interpreting the results of the experiments. In particular, were the urchins fed on seaweed prior to the feeding trials? If so, which seaweed(s)? Were the smallest urchins fed the same diet as the largest urchins? Were the urchins starved prior to the feeding trials?

The experimental design and methods for the growth trials on the algae singly are consistent with similar studies on other sea urchins. Feeding the algae ad libitum is likely to have avoided resource competition amongst the three size classes.

The experimental design and methods for the no-choice feeding experiment are consistent with similar studies on sea urchins.

The experimental design and methods for the choice feeding experiments are not consistent with similar studies on sea urchins. In particular, the use of four sea urchins in each replicate instead of a single urchin raises questions about the ability of the sea urchins to choose between the seaweeds. For example, if 1-3 sea urchins respectively were feeding on a particular seaweed, did this prevent access to that particular seaweed by the remaining urchins? If so, this may have led to higher feeding rates on ‘less preferred’ seaweeds than would have been the case if only one urchin was used in each replicate.

I am concerned about the lack of replication in the experimental design for the field caging experiment, where n = 2 for each cage type. I would avoid publishing this data, but I will leave this to the judgment of the editor. Some journals are happy to publish studies where n = 2, whereas others require n to be three or greater.

Validity of the findings

The authors need to carefully consider presenting the growth rates of the three size classes of the urchins fed on the single species of seaweed separately for two reasons. Firstly, growth was calculated as growth rates. Smaller urchins have inherently higher growth rates than larger urchins. This is especially true when measuring test diameter as an increase in test diameter for large urchins requires a substantial increase in volume, but less so for small urchins. Although initial size was accounted for using ANCOVA, impaired growth of the smallest urchins in some treatments could skew the data irrespective of whether growth of the largest urchins was impaired (i.e. larger effect of diet on growth of smaller urchins versus large urchins). No indication is given as to whether this occurred. If the differences in growth rates amongst the diets were consistent across the size classes, this should be stated in the results. Secondly presenting this data would be useful for readers who may wish to compare growth rates of the sea urchins between this and other studies.

Additional comments

The authors should revise the manuscript carefully to remove grammatical errors. For example, Ln 225 “though not significantly lower to urchins on diets of A. spicifera”. A. spicifera is one diet, not multiple diets. Ln 211 and 248: “3” should be written as ‘three’.

The manuscript could be substantially shortened, particularly by the removal of informal language which appears in the Introduction and Discussion. The manuscript also contains sentences which are too long, particularly in the Introduction and Discussion, which makes reading difficult.

The subheadings within the Methods and Results need to be consistent. For example Ln 150 No-Choice Feeding Preference versus Ln 227 No-choice Feeding Trial. The descriptions used also need to be carefully considered. For example, if the urchins were only offered one seaweed, logically they could not show a preference.

Ln 238: “feeding rates were the same as the medium urchins,” This is confusing. Did the small urchins eat the same amount (weight) as the medium urchins? Or were the same algae preferred?

Reviewer 2 ·

Basic reporting

.

Experimental design

.

Validity of the findings

.

Additional comments

General comments

This paper should be considered for publication. I have a few comments but think that this is an important piece of work that contributes to our understanding of biotic control.

The results are quite interesting and discussed with a focus of potential management actions. The fact that each size classes, except for the biggest size, responded different to the species that were offered helps define a more efficient management strategy taking into account body size

This is a clear example of how science can be applied to management.

INTRODUCTION

The introduction is rather long, some suggestions to shorten it:

Line 26, 27 and 28 could be omitted or shortened by just stating

The information about the origin of the introduced species should be shortened or send to the appendix to make the intro shorter and more focused on biotic control which is a central topic of this paper. For example, a review of the suitability of other marine taxa as biotic control agents is missing and could emphasize the role of sea urchins as such agents.


There is no defined hypothesis in the introduction

A recent review about biotic control by Kimbro et al 2013 published in Ecology Letters should be included as this meta analysis brings a lot of information about this topic with respect to the role of biotic interactions as part of the control of invasive species, particularly with respect to type of interaction (competition, vs. consumption and both), latitude and type of community (i.e. rocky shores, sandy beaches, subtidal reefs, among others).


Also, it will be interesting to know what is the relationship between the abundance of this sea urchin and the percent cover of the different types of invasive algae.


METHODS SECTION

In line 107 and 108 the authors mention that this sea urchin is relatively common, however, they don’t mention their current abundance.

The fact that the sea urchin is not very abundant in calm waters makes me wander about their biotic control in this type of environmental conditions where one species of algae seems to flourish. this also happens with Caulerpa in
Galapagos.

LINE 127, 128 AND 129 COULD BE OMMITED AND LEFT FOR THE AKNOWLEDGEMENT SECTION

It is not clear what was the level of replication at the site. It seems according to the method section that there were only two replicates per site, it seems to me that at least 5 replicates per treatment per site should have been included in the analysis to give more power to the results.

A procedural control was not used in this experiment; I wander if they have any evidence of artifact effects that resulted from the use of their experimental units.

Also, the authors did not mentioned if the data comply with normality or variance homogeneity.


RESULTS

Growth on Single-species Diet: In the text there is no reference to the figure that shows the results of this section of the experiments

LEGENDS

Some of the legends need more detail about the experiment so they can be self contained. For example, there is no indication of the location of the experiments. Additionally, mentioning the level of replication in each one of the legends will be useful for the reader.

---

## Round 0.2 · accepted · Accept

Authors have made all the suggestions indicated and the manuscript has been improved.